# In Vitro Cell Proliferation and Migration Properties of Oral Mucosal Fibroblasts: A Comparative Study on the Effects of Cord Blood- and Peripheral Blood-Platelet Lysate

**DOI:** 10.3390/ijms24065775

**Published:** 2023-03-17

**Authors:** Arief Faisal Azmi, Mohammad Amirul Asyraff Mohd Yahya, Nur Ain Azhar, Norliwati Ibrahim, Norzana Abd Ghafar, Nur Azurah Abdul Ghani, Muhammad Aiman Mohd Nizar, Siti Salmiah Mohd Yunus, Tashveender Kaur Lakhbir Singh, Jia-Xian Law, Sook-Luan Ng

**Affiliations:** 1Department of Craniofacial Diagnostics and Biosciences, Faculty of Dentistry, Universiti Kebangsaan Malaysia, Jalan Raja Muda Abdul Aziz, Kuala Lumpur 50300, Malaysia; 2Department of Anatomy, Faculty of Medicine, Universiti Kebangsaan Malaysia, Jalan Yaacob Latif, Bandar Tun Razak, Cheras, Kuala Lumpur 56000, Malaysia; 3Department of Obstetrics and Gynaecology, Faculty of Medicine, Universiti Kebangsaan Malaysia, Jalan Yaacob Latif, Bandar Tun Razak, Cheras, Kuala Lumpur 56000, Malaysia; 4Department of Oral and Maxillofacial Surgery, Faculty of Dentistry, Universiti Kebangsaan Malaysia, Jalan Raja Muda Abdul Aziz, Kuala Lumpur 50300, Malaysia; 5Centre for Tissue Engineering and Regenerative Medicine, Faculty of Medicine, Universiti Kebangsaan Malaysia, Jalan Yaacob Latif, Bandar Tun Razak, Cheras, Kuala Lumpur 56000, Malaysia

**Keywords:** cord blood, peripheral blood, platelet lysate, oral ulcer, wound-healing

## Abstract

Cord blood-platelet lysate (CB-PL), containing growth factors such as a platelet-derived growth factor, has a similar efficacy to peripheral blood-platelet lysate (PB-PL) in initiating cell growth and differentiation, which makes it a unique alternative to be implemented into oral ulceration healing. This research study aimed to compare the effectiveness of CB-PL and PB-PL in promoting oral wound closure in vitro. Alamar blue assay was used to determine the optimal concentration of CB-PL and PB-PL in enhancing the proliferation of human oral mucosal fibroblasts (HOMF). The percentage of wound closure was measured using the wound-healing assay for CB-PL and PB-PL at the optimal concentration of 1.25% and 0.3125%, respectively. The gene expressions of cell phenotypic makers (Col. I, Col. III, elastin and fibronectin) were determined via qRT-PCR. The concentrations of PDGF-BB were quantified using ELISA. We found that CB-PL was as effective as PB-PL in promoting wound-healing and both PL were more effective compared to the control (CTRL) group in accelerating the cell migration in the wound-healing assay. The gene expressions of Col. III and fibronectin were significantly higher in PB-PL compared to CB-PL. The PDGF-BB concentration of PB-PL was the highest and it decreased after the wound closed on day 3. Therefore, we concluded that PL from both sources can be a beneficial treatment for wound-healing, but PB-PL showed the most promising wound-healing properties in this study.

## 1. Introduction

Generally, an oral ulcer recovers quickly and efficiently. However, if the oral ulcerative lesion continues for more than a week or longer, it is indicated as a chronic ulcer. Ulcerations can be defined as defects in the epithelium, connective tissue or both. Intra-orally, ulcerative lesions can be divided into three major categories, which are acute, chronic or recurrent ulcers, which are then further characterized into five subgroups: solitary acute, multiple acute, solitary chronic, multiple chronic and solitary/multiple recurrent. It is categorized based on the numbers and durations of ulcer lesions [1]. Oral lichen planus (OLP) is a chronic inflammatory, mucocutaneous, autoimmune disease which is quite common in the female oral mucosa. Erosive OLP is characterized by ulceration and erosion in oral mucosa and causes severe pain in patients [2]. The recommended topical therapies for erosive OLP are topical corticosteroids, tacrolimus and photodynamic therapy. Oral corticosteroid is the first line systemic therapy for erosive OLP, but its prolonged use can cause side effects such as weight gain, muscle weakness, sleep disorders, etc.

Wound-healing is a four-phase process that involves hemostasis, inflammation, proliferation and remodeling to restore tissue integrity following trauma. Different cell types and signaling molecules are uniquely involved in each phase [3]. The human oral mucosal fibroblast (HOMF), which is derived from the mesenchymal cells in connective tissue, plays a unique role in the extracellular matrix (ECM) synthesis and organization. Under normal circumstances, it plays a critical function in maintaining ECM homeostasis. Fibroblasts produce type 1 collagen (Col. I) in the remodeling phase to reduce scarring [4]. The main role of Col. I is to maintain the structural and biological ECM integrity. It undergoes frequent remodeling to clarify the cellular action as well as tissue activity due to its flexibility. It is surface-active and able to penetrate the lipid-free interface [5]. Col. I is commonly associated with Col. III to contribute the tissue mechanical strength and flexibility as well as to serve as a platform for cell adhesion, proliferation and differentiation [6,7]. Fibronectin is important to facilitate the assembly of elastic fibers and improve the elasticity and mechanical properties in ECM remodeling [8].

Platelet-rich plasma (PRP) is a component produced from autologous platelet blood. It is high in growth factors, cytokines and chemokines. It contains fundamental growth factors, such as the vascular endothelial growth factor (VEGF), transforming growth factor-beta 1 (TGF-β1), insulin-like growth factor (IGF) and platelet-derived growth factor (PDGF), all of which promote wound repair and regeneration [9]. A review carried out by Albanese and colleagues showed that PRP has many positive healing effects in post-extraction, post-periodontal surgery, and post-implant surgery [10]. The PRP application also exhibited a decreased closing time of the ulcers in patients with Behcet’s disease [11]. In addition, another hemoderivative material, platelet lysate (PL) was observed to be effective in wound and ulcer healing, including oral mucositis [12]. PL is obtained by the repeated freezing-thawing of the PRP [9].

Cord blood (CB) plasma collected from human umbilical cord blood was previously considered a waste product and was commonly discarded [13]. This is because 80% of accepted CB units have a low amount of total nucleated cells and a small volume (less than 120 mL). Thus, it was not used for further procedures and was discarded. PRP is mostly harvested from adult peripheral blood (PB), but recently, CB has been proposed as a remarkable alternative to preparing PL [14]. Human cord blood platelet lysate (CB-PL) was proven to have higher levels of pro-angiogenic growth factors compared to peripheral blood platelet-rich plasma (PB-PRP) [15]. The presence of growth factors from both sources was tested and there were significantly high levels of epidermal growth factor (EGF), fibroblast growth factor (FGF), transforming growth factor-alpha (TGF-α), transforming growth factor-beta 2 (TGF-β2), PDGF, VEGF and nerve growth factor in CB samples in comparison to PB samples. In contrast, IGF-1, IGF-2 and TGF-β1 levels are higher in PB samples [16].

The concentration of PL has been proven to play an important role in determining the wound-healing rate. Cell viability and proliferation improved with PL concentrations of 10% and 20% compared to the lower concentrations. The wound-healing rate improved as concentration was increased [17]. However, Chen and colleagues showed that PL at lower concentration has higher cell proliferation through the MTS (3-(4,5-dimethylthiazol-2-yl)-5-(3-carboxymethoxyphenyl)-2-(4-sulfophenyl)-2H-tetrazolium) assay [18]. This finding was consistent with a study by Losi et al. that showed lower concentrations of CB-PL, such as 1%, 3% and 5%, of the total medium volume, resulted in increased cell viability and proliferation, and was inhibited in higher concentrations [15].

Many studies have shown that PL could be used as an adjunct treatment for wound-healing. CB-PL exhibits a higher mitogenic effect on corneal epithelial cells and thus it could be used as an alternative eye drop to promote corneal wound-healing in patients with Dry Eye Syndrome [19]. Growth factor-rich CB-PL is potentially used as therapeutic agent to heal chronic wounds [20]. On the other hand, PB-PL can be utilized as an alternative to FBS to expand the chondrocytes by maintaining their chondrogenic features for the purpose of autologous chondrocyte implantation [21,22]. PB-PL is also effective for promoting skin wound-healing in both in vitro and in vivo models [23]. However, different types of blood should be considered, as findings showed that cord blood heals wounds better than peripheral blood. The optimal concentration must also be considered, as it may have a significant impact on the healing process. As a result, the objectives of this study are to determine the appropriate concentrations of CB-PL and PB-PL in wound-healing, as well as to compare the healing rates and properties of CB-PL and PB-PL.

## 2. Results

### 2.1. Optimal Concentrations of CB-PL and PB-PL in Cell Proliferation Assays

The cell proliferation of HOMF treated with an increasing concentration of CB-PL and PB-PL was evaluated with Alamar blue assay, and the results are shown in Figure 1. Alamar blue proliferation assay was performed in order to determine the optimal concentration of CB-PL and PB-PL. the highest value of reduction indicated the highest amount of cell proliferation. In Figure 1, the culture media mixed with 0.3125% PB-PL and 1.25% of CB-PL showed the highest HOMF cell proliferation. These results indicated the optimal concentration of each PL. Serum-free medium was used as negative control to show the dose–response relationship of PL on cell proliferation [17]. The cell proliferation showed an upward trend with the increase in PL concentration. However, the cell proliferation started to decrease after it peaked at 0.3125% for PB-PL and at 1.25% for CB-PL. A Kruskal–Wallis test showed that CBPL significantly affects the cell proliferation of HOMF, *H* (7) = 31.788, *p* < 0.001. The cell proliferation of CBPL 1.25% (*Mdn* = 110.48) was higher than CTRL (0%) (*Mdn* = 100.00). A Mann–Whitney test indicated that this difference was statistically significant, *U* (*N*_CTRL_ = 6, *N*_CBPL1.25%_ = 6) = 0.000, z = −3.077, *p* = 0.002. Moreover, a Kruskal–Wallis test also showed that PBPL significantly affects the cell proliferation of HOMF, *H* (7) = 43.881, *p* < 0.001. The cell proliferation of PBPL 0.3125% (*Mdn* = 119.00) was higher than the CTRL (0%) (*Mdn* = 100.00). A Mann–Whitney test indicated that this difference was statistically significant, *U* (*N*_CTRL_ = 6, *N*_PBPL0.3125%_ = 6) = 0.000, z = −3.077, *p* = 0.002. The cell proliferation of PBPL 5% (*Mdn* = 71.80) was lower than the CTRL (0%) (*Mdn* = 100.00). A Mann–Whitney test indicated that this difference was statistically significant, *U* (*N*_CTRL_ = 6, *N*_PBPL5%_ = 6) = 0.000, z = −3.077, *p* = 0.002. There was no statistically significant difference in cell proliferation between other concentrations when compared to CTRL.

### 2.2. Wound Closure Percentage in Wound-Healing Assay

The wound closure percentage between CP-PL, PB-PL and CTRL were compared by using the wound-healing assay. The addition of CB-PL and PB-PL increased the migration of HOMF cells compared to CTRL as demonstrated in Figure 2, Figure 3 and Appendix A. There was a statistically significant difference between three groups (CTRL, 1.25% CP-BL and 0.3125% PB-PL) as determined by one-way ANOVA for the wound closure percentage at 24 h and 72 h. A Tukey post hoc test revealed that the wound closure percentage of 1.25% CB-PL (43.81 ± 6.13%, *p* = 0.015) was higher than that of the CTRL (22.22 ± 0.54%) at 24 h. There was no statistically significant difference in wound closure percentage when 0.3125% PB-PL (31.20 ± 2.08%, *p* = 0.282) was compared to CTRL at 24 h. On the other hand, the wound closure percentage of 1.25% CB-PL (78.80 ± 6.32%, *p* = 0.012) and 0.3125% PB-PL (74.22 ± 3.31%, *p* = 0.030) were higher than that of the CTRL (53.29 ± 1.64%) at 72 h. Although there was no significant difference between CB-PL and PB-PL, both PLs were still comparable and better than the CTRL. The optimal concentration that was used for both CB-PL and PB-PL were 1.25% and 0.3125%, respectively, based on the Alamar blue assay results that are shown in Figure 1.

### 2.3. Relative mRNA Expressions of Cell Phenotypic Markers in HOMF

There was a statistically difference between groups as determined by one-way ANOVA for the relative mRNA expression of Col. III and fibronectin (Figure 4). However, there were no differences between the three groups (CTRL, 1.25% CP-BL and 0.3125% PB-PL) in the relative mRNA expression of elastin. A Tukey post hoc test revealed that the relative mRNA expressions of both Col. III and fibronectin were higher with the treatment of 0.3125% PB-PL (4.62 × 10^−1^ ± 3.50 × 10^−2^, *p* = 0.02; 2.57 ± 2.83 × 10^−1^, *p* = 0.037) compared to 1.25% CB-PL (2.49 × 10^−1^ ± 5.43 × 10^−2^; 1.56 ± 2.15 × 10^−1^). A Kruskal–Wallis test showed no significant difference between three groups (CTRL, 1.25% CP-BL, and 0.3125% PB-PL) in their Col. I gene expressions, *H*(2) = 5.956, *p* = 0.051 with median values of CTRL (*Mdn* = 3.12), 1.25% CB-PL (*Mdn* = 2.08) and 0.3125% PB-PL (*Mdn* = 3.68).

### 2.4. Human PDGF-BB Measurement

PDGF-BB concentrations of 100% CB-PL and 100% PB-PL before use for testing were 35,250.71 ± 395.80 pg/mL and 53,015 ± 741.65 pg/mL, respectively (Appendix A). PDGF concentration was measured in control in order to know the baseline of PDGF concentration in HOMF during wound-healing. Fibroblasts are known to secrete PDGFs in response to injury [24]. A one-way ANOVA was performed to compare the PDGF-BB concentration between the three different groups (CTRL, 1.25% CP-BL, and 0.3125% PB-PL) on day 0 and day 3 (Figure 5), where there was a statistically significant difference in PDGF-BB concentration between three groups on day 0 and day 3. Tukey’s HSD test for multiple comparisons found that 1.25% CB-PL and 0.3125% PB-PL had higher PDGF-BB concentrations (876.19 ± 5.30 pg/mL, *p* < 0.001; 1065.24 ± 11.56 pg/mL, *p* < 0.001) as compared to the CTRL (588.57 ± 12.21 pg/mL) on day 0. In addition, PDGF-BB concentrations of 1.25% CB-PL and 0.3125% PB-PL (876.19 ± 5.30 pg/mL, *p* < 0.001; 1065.24 ± 11.56 pg/mL, *p* < 0.001) were also higher than CTRL (588.57 ± 12.21 pg/mL) on day 3. The PDGF concentration of 0.3125% PB-PL was significantly higher than 1.25% CB-PL on day 0 (*p* ≤ 0.001) and day 3 (*p* = 0.025). Independent *t*-tests were performed to compare the PDGF-BB concentrations between day 0 and day 3 for each group. It was found that PDGF-BB concentrations of both 1.25% CB-PL and 0.3125% PB-PL on day 3 were significantly decreased as compared to day 0 (*p* < 0.001).

## 3. Discussion

The principal finding of our study is that CB-PL is as potentially useful as PB-PL in oral wound-healing, and our research revealed that CB-PL could be a safe option for the treatment of oral ulcerations. Cord blood, when compared to peripheral blood, showed a higher level of viral safety; has higher levels of many types of growth factors [15,16], particularly VEGF and PDGF [25]; and skin wounds for blood collection during CB-PL preparation is avoidable. Studies have shown the benefits of a variety of platelet therapies, particularly PRP, in the treatment of oral ulcerations. Common ulcers, such as oral lichen planus [26], pemphigus vulgaris [27] and oral mucositis [28], as well as less common lesions such as oral ulcers in Behcet’s disease [11], were treated with mucoadhesive, rinses or intralesional injections of autologous PRP, which proved to reduce ulcerative pain and thus improved mastication. Zamani and colleagues used platelet lysate to treat oral mucositis [12], and the research team of Sindici showed evidence of oral ulcer improvement in epidermolysis bullosa patients treated with cord blood platelet gel [29]. Although these platelet therapies differ in their methods of preparations and sources (using cord or peripheral blood), the therapeutic potential of platelets, which are safe and easy to use, was proven. In order to obtain the PL, both collected bloods underwent double centrifugation to produce the PRP, followed by three freezing and thawing cycles [9]. After PL was obtained, it was added to seeded cells in 96-well plate and tested with the Alamar blue assay to find out the optimal concentration. The highest percentage in the reduction of absorbance reading indicates the highest number of viable cells. Next, a wound-healing assay was performed to compare the healing rates of both PL.

### 3.1. Limitations and Advantages of PB-PL versus CB-PL

Traditionally, autologous peripheral blood has been used, but its application is limited, as is it can be impractical due to the fact that it can be hard to maintain the amount of blood needed to sustain the supply of PL. Significant and practical limitations, such as the need for repeated blood collection, might be difficult or clinically inappropriate for some different categories of patients, such as elderly hypomobile patients, neonates, and children [30]. In recent studies, it has been shown that CB can be an alternative source of PL, as it is rich in growth factors. The research group of Losi demonstrated that CB-PL has high concentrations of PDGF-AB, TGF-β1 and VEGF in respect to PB-PL, which can influence the wound-healing rate [15].

To obtain the PL, the collected blood underwent double centrifugation to produce the PRP and was followed by three freezing and thawing cycles [9]. After PL was obtained, it was added to a seeded cells in 96-well plate and Alamar blue assay was performed to find out the optimal concentration. The highest percentage in the reduction of absorbance reading indicates the highest number of viable cells. Next, a wound-healing assay was carried out to compare the healing rates of both PLs.

In preparing the PL, the platelet count of each PRP was found to be at the range between 200–400 × 10^9^ platelets/L. In comparison to a working definition of PRP described by Marx [31], our platelet count was much higher. This may be related to the optimal concentrations of CB-PL and PB-PL found using Alamar blue assay, which was 1.25% and 0.3125%, respectively. These concentrations were actually lower when compared to other studies. Losi and colleagues showed an optimal concentration of 10% in CB-PL [15], while Barsotti et al. demonstrated an optimal concentration of 20% for PB-PL [17]. The difference in the optimal concentration may be due to the different levels of growth factors in each PL [15].

Moreover, Hassan et al. (2021) showed that 10% PB-PL significantly increased the cell number and reduced the population doubling time compared to the complete medium with 10% FBS [32]. Barsotti et al. (2013) reported that scratch closure of 10% PB-PL was comparable to the 10% FBS group [17]. On the other hand, our results showed a significant decrease in cell proliferation (<80%) when the cells were treated with 5% PB-PL. Similarly, Losi et al. (2020) reported that a high concentration of PL could inhibit cell viability and proliferation due to the high amount of TGF-β1 in PL, which induces cell apoptosis [15]. A lower concentration of PB-PL (0.3125%) promoted a better cell proliferation on HOMF. This contradiction may be due to differences in PL preparation method, which resulted in PL with different platelet counts and growth factor concentrations. Although 5% and 10% PB-PL increased the cell proliferation, the cell migration and ECM gene expression were decreased compared to 10% FBS, due to the addition of heparin as an anti-coagulant in the culture medium [32]. Our findings also showed that ECM gene expressions were downregulated after being treated with CB-PL and PB-PL. In contrast, the cell migration and wound-healing were increased in conjunction with the increase in cell proliferation. Similarly, Sergeeva et al. (2016) found that PL promoted wound closure in vitro [23]. In the wound-healing assay, CB-PL showed no significant difference to PB-PL. This proved that CB could be an alternative source for PL, as it is as effective as PB in enhancing wound-healing. Both PB-PL and CB-PL showed a higher rate of wound-healing compared to the control. Thus, PL from both sources can be an effective treatment for wound-healing. This same result in seen in other studies [15,17].

### 3.2. The Role of Platelet in Wound-Healing and the Biomarkers

Platelets in our blood contain growth factors that are important for body’s local wound-healing process, in which it can stimulate a repair response. Platelets orchestrate this response by releasing multiple growth factors one after another within one week. These include the release of EGF, PDGF, VEGF, FGF, TGF-β and the keratinocyte growth factor. One of the means of improving the healing response is to concentrate these platelets in the blood by centrifugation, and this creates platelet-rich plasma, which can be used in medical treatment, for example, for mild arthritis. In addition, accelerate healing response can be achieved through the lysis of the platelets, which rapidly release alpha granules containing growth factors, and this is called platelet lysate. PL study has attracted many researchers for a new therapeutic strategy in regenerative medicine [12,33].

Platelets release PDGF instantaneously after injury and attracts inflammatory cells, such as neutrophils, monocytes and fibroblasts. The main role of PDGF in platelets is to accelerate the migratory process of these cells towards the site of injury, which in turn encourages wound-healing processes, such as re-epithelialization, mitogenic activity and angiogenesis. Other growth factors of platelets, such as EGF, cause keratinocytes to migrate toward injury areas, while VEGF and TGF-β promote angiogenesis and healing in connective tissues. Platelets also play a crucial role as antimicrobial agents against *Staphylococcus aureus* and *Escherichia coli*. They produce mediators such as hydroxyl free radicals, superoxide and hydrogen peroxide to combat bacterial wound infection that delays healing. Apart from growth factors, platelets also have complementing proteins that inhibit bacteria progression at wound sites [33].

Parazzi and colleagues suggested that cord blood platelet gel had higher levels of VEGF and PDGF compared to adult peripheral blood, and this may be desirable in skin and oral mucosal wound-healing [25]. VEGF is a key molecule in the promotion of angiogenesis and neovascularization, while PDGF is crucial in the proliferation of mesenchymal stem cells. In contrast to the literature, this study found that PB-PL contains higher PDGF-BB than CB-PL, although platelet count was adjusted to be the same for both PL. This is likely the reason why the optimal concentration of PB-PL for cell proliferation and wound-healing is lower than CB-PL. Conversely, the research group of Hashemi revealed that no significant difference was observed between cord blood and adult peripheral blood PRP preparations concerning cell proliferation and migration [34]. Subsequently, they proposed that, in circumstances where adult PRP may not be accessible, umbilical cord blood PRP could be utilized with the assumption that it would deliver comparable results of cell proliferation and migration required in wound-healing.

This study showed that the mRNA expression of Col. I was higher than Co1. III after the wound closed, as Col. I synthesis normally increased during the late stage of wound-healing, while Col. III was synthesized more during the early stage and then replaced by Col. I [6]. PB-PL treatment contributed to a higher mRNA ratio of Col. I to Col. III when compared to CB-PL in this study. A high ratio of Col. I/Col. III was found in hypertrophic scars and keloids [35]. However, the research group of Wan reported different results concerning the ratio of collagens, where the ratio of Col. I/Col. III was proportional to the mechanical strength of the connective tissue [36]. On the other hand, our findings were similar to the study of Sá and colleagues, where PDGF production was reduced with the increase in Col. I and Col. III gene expression after wound closure [37]. The remodeling of Col. I to Col. III fibers was postulated. Overall, our study showed that PB-PL contained higher concentrations of PDGF-BB and the promoted higher gene expression of Col. I, Col. III, fibronectin and elastin than CB-PL. Fibronectin was upregulated with the addition of PDGF-BB during cell expansion [38]. Fibronectin and elastin also worked well together to enhance cell proliferation and the Col. I gene expression of human mesenchymal stem cells [39].

### 3.3. Possible Reasons for the Low Platelet Count in Subjects and Risk Reduction Activities

Our subjects in this study showed a lower platelet count, as compared to other studies, which in turn affects the optimal concentration. Thus, here we revise the possible reasons for the low platelet count and the adequacy of the risk reduction activities of the samples. The first step of evaluation is the blood donors themselves, by excluding donors who have known illnesses or who may transfer infectious diseases or are pregnant. As the possible etiologies of low platelet count or thrombocytopenia are abundant, re-evaluating the previous blood test findings of donors is necessary. A patient’s low platelet count can be typical for them and not indicative of sickness. Reference ranges would, by definition, only include 95% of the population. In addition, it is safe to monitor a patient with a stable/asymptomatic condition and a mildly low platelet count (50–100 × 10^9^/L) who has no additional cytopenia [40]. In addition, the current COVID-19 pandemic situation may have played a role in influencing the low platelet count in our subjects. A single-center retrospective observational study evaluates the platelet trends and clinical outcomes of 34 patients with chronic or persistent immune thrombocytopenia (ITP) after receiving a two-dose vaccination series against COVID-19. Platelet counts before vaccination (baseline) to 6 weeks after the completion of the vaccination series were recorded. Result showed 44.1% had a decrease, 38.2% had no change and 17.6% had an increase after the second dose of the COVID-19 vaccination. They also demonstrated that platelet decreases were mostly transient [41]. Our subjects, who had been vaccinated with two or three doses, may have similarly experienced a transient low platelet count, as observed in that study. Nevertheless, the impact of vaccination in people with a history of ITP is little understood.

### 3.4. Future Studies

The proteomic profiles of PL can be evaluated by mass spectrometry to identify the amount and type of growth factors during PL characterization [42]. In addition, cell doubling time can be performed together with cell proliferation to well define the cell growth. The differentiation capability of cells in relation to PL can also be studied in the future studies. For example, the angiogenic potential of CB-PL and PB-PL can be studied as PL contains pro-angiogenic factors, such as PDGF, which is known to promote blood vessel formation during wound-healing. Fibroblasts were able to differentiate into endothelial-like cells and promote tubular formation after treated with PL.

## 4. Materials and Methods

### 4.1. Sample Collection and Preparation

There were three types of samples collected for this study, namely (i) cord blood, (ii) peripheral blood and (iii) oral mucosal tissue. The first specimen, the cord blood, was harvested by an obstetrician from the Department of Obstetrics and Gynaecology, Universiti Kebangsaan Medical Centre (UKMMC). An amount of 150–200 mL of cord blood was collected from healthy newborns (*n* = 6) weighing 3.0–3.5 kg in blood bags containing a citrate-phosphate-dextrose-adenine anticoagulant [43]. Secondly, human peripheral blood (30 mL) was collected from a healthy donor by a phlebotomist (*n* = 6) at the Department of Anatomy, UKMMC. Lastly, the primary human oral mucosal fibroblasts (*n* = 6) were obtained from normal oral mucosal tissues. These normal tissues (approximately 1 or 2 mm diameter of excess normal tissues) were gained either during wisdom tooth surgical removal or through oral mucosal lesion biopsy. The surgeries were carried out at the Oral Surgery Clinic or Oral Medicine Clinic, Faculty of Dentistry UKM, respectively. The patients were duly informed about the study and information sheets with informed consent forms were provided.

The inclusion criteria for the cord blood involved healthy newborns with no infectious diseases, confirmed through the screening of the maternal blood for HIV, HBV, hypertension and diabetes mellitus. Exclusion criteria were subjects that were not healthy and were not free of infectious diseases [44]. The inclusion criteria for the peripheral blood were that the donor must be a healthy adult of a minimum age of 21 years old and with no history of tobacco smoking, hypertension, diabetes mellitus or any critical illness. The exclusion criteria involved the donor being underweight (<45 kg) and not having had enough sleep (<5 h). For HOMF cells, the tissue specimen was obtained from the normal oral mucosal of the oral cavity of patients who were healthy with no soft tissue lesions [45] as well as no history of drug intake in the past six months.

The following in vitro procedures took place at the Tissue Culture Laboratory, Faculty of Dentistry, UKM.

#### 4.1.1. Cord Blood Collection

Once the baby was delivered and assessed, the obstetrician clamped and cut the cord to start collecting the blood immediately, while the placenta was still inside the uterus. A povidone–iodine solution was used to clean the site of blood collection [46]. A 16-gauge needle that was connected to a sterile bag containing anticoagulant citrate, nutrient phosphate and dextrose (JMS, Singapore) punctured the umbilical vein. Blood was then collected in the bag by the force of gravity. A total of approximately 150–200 mL of cord blood could be collected per procedure. This procedure had lower macroscopic clots and did not interrupt the natural course of birth or the postpartum period. The processing of the blood should be performed within 24 h after the collection. The Fact-Netcord standard states that CB collections must be limited to uncomplicated deliveries and in utero CBU collections can only be obtained from infant donors that exceed 34 weeks gestation [47].

#### 4.1.2. Peripheral Blood Collection

Peripheral blood was collected by a phlebotomist. The donor sat with their arm extended on an armrest. The upper arm was wrapped with a tourniquet to allow the collection of blood from the veins. The skin, commonly on the inside of the elbow, was wiped with alcohol swabs. Following that, a sterile 21-gauge needle with a syringe was inserted into the vein. A total of ten 3ml-K2EDTA tubes (BD, Franklin Lakes, NJ, USA) of blood were collected from a donor. After the collection was completed, the needle was removed and a cotton ball was placed on the puncture site followed by an application of pressure to stop any bleeding. It was necessary to perform the next procedure within 24 h after blood collection.

#### 4.1.3. Blood Processing

Cord blood units were collected into sterile bags containing anticoagulant citrate, nutrient phosphate and dextrose, whereas peripheral blood was collected into the EDTA tubes. Using a sterile syringe with an 18-gauge needle, the cord blood sample from the blood bag was transferred to a 10 mL sterile centrifuge tube (SPL, Pocheon-si, Republic of Korea) inside a biosafety cabinet class II. The centrifugation of the cord blood was performed at 300× *g* for 25 min at room temperature (24 °C). A sterile pipette was inserted above the buffy coat to remove the PRP and transferred to a new sterile centrifuge tube. A second centrifugation was carried out at 600× *g* for 5 min at room temperature. Next, ⅔ of the centrifuged plasma (supernatant) was removed and transferred into a sterile centrifuge tube using a sterile pipette. PRP collected at the bottom of the tube was resuspended in the remaining plasma. Centrifugation for peripheral blood was performed in the same way except that the first centrifugation was carried out at 100× *g* for 15 min, while the second one was performed at 600× *g* for 5 min [48]. Both blood samples were then sent to the Pathology unit, UKMMC, for the analysis of platelet count using a SYSMEX Haematological Analyser (SYSMEX, Tokyo, Japan). The total platelet count of platelet-rich plasma derived from umbilical cord blood (UCB-PRP) and peripheral blood (PB-PRP) were 378 ± 59^9^/L and 486 ± 50^9^/L, respectively (Appendix A). The PRP was adjusted to the platelet count of 300 × 10^9^/L. Platelet lysate for both CB and PB were obtained after 3 cycles of the freeze–thaw process, followed by centrifugation carried out at 5000× *g* for 20 min at room temperature. The CB-PL and PB-PL were pooled after the centrifugation was finished. PL was kept at −80 °C for further use.

#### 4.1.4. Primary Human Oral Mucosal Fibroblast Isolation and Culture

The primary HOMF were isolated using the enzymatic method. The soft tissue specimens were stored in phosphate-buffered saline supplemented with 1% antibiotic/antimycotic (Gibco-BRL, Waltham, MA, USA). In order to dissociate the cells from the tissue matrix, the soft tissue was digested by 5 ml 0.3% collagenase type I (Worthington Biochemical Corporation, Lakewood, NJ, USA) in a 37 °C shaker incubator for 15 min. The cells were cultured in one T25 culture flask (SPL, Republic of Korea) using Dulbecco’s Modified Eagle’s Medium/Nutrient Mixture F-12 (DMEM/F-12, Gibco-BRL, Paisley, UK) supplemented with 10% fetal bovine serum (FBS, Gibco-BRL), 1% antibiotic/antimycotic (Gibco-BRL, USA), 1% L-glutamine (Gibco, Tokyo, Japan) and 1% L-ascorbic acid (HmbG Chemicals, Germany) in a 37 °C incubator with 5% CO_2_ atmosphere. The culture medium was replaced every 2 to 3 days. After 7–10 days, the cells were cultured until 70–80% confluence and underwent quick trypsinization to harvest the pure fibroblasts and leave the epithelial cells in the culture flask. Quick trypsinization was performed with incubated the cells through 2 mL 0.05% trypsin-EDTA (Gibco-BRL, USA) at 37 °C for 5 min [49]. Passage 2 HOMF was harvested and used for cell characterization via cell morphology observation and the mRNA expression of vimentin (fibroblast marker). HOMF showed fibroblastic features, such as a flat and spindle shape (Appendix A) and expressed vimetin (Appendix A). After that, HOMF were cultured until passage 4, involving Alamar blue assay and wound-healing assay.

### 4.2. Alamar Blue Cell Proliferation Assay

In order to determine the optimal concentration of cord blood and peripheral blood platelet lysate, Alamar blue assay was used to measure the cell proliferation of HOMF. In the first step, the oral mucosal fibroblasts with a seeding density of 5 × 10^3^ cells per well were seeded onto the 96-well plate using 0.1 mL culture medium. After 24 h of incubation, the medium was carefully removed and replaced with 0.1 ml serum-free medium (DMEM/F-12, 1% antibiotic/antimycotic, 1% L-glutamine and 1% L-ascorbic acid) containing 0%, 0.078%, 0.156%, 0.313%, 0.625%, 1.25%, 2.5% and 5% (*v*/*v*) CB-PL and PB-PL, respectively. The control (CTRL) group comprised cells treated with serum free medium (0% PL). Cells were allowed to grow for another 72 h at 37 °C. After the 72 h incubation period, 10 µL of culture medium was removed and replaced with 10 µL Alamar blue reagent (Thermo Fisher Scientific, Waltham, MA, USA) and cultures were incubated at 37 °C for 4 h. At the end of the incubation time, the optical density was measured at a wavelength of 570 nm and 600 nm. The percentage of reduction value of Alamar blue was calculated based on the manufacturer’s protocol. Subsequently, the percentage of cell proliferation was obtained using the equation as follows:(1)$$\%\;\mathrm{Cell}\;\mathrm{Proliferation}=(\frac{\%\;\mathrm{Reduction}\;{\mathrm{Value}}_{\mathrm{test}}-\%\;\mathrm{Reduction}\;{\mathrm{Value}}_{\mathrm{CTRL}}}{\%\;\mathrm{Reduction}\;{\mathrm{Value}}_{\mathrm{CTRL}}})\times 100\%$$

### 4.3. Wound-Healing Assay

A wound-healing assay was used to determine the percentage of wound closure for both PLs [50]. Firstly, HOMFs were cultured in 6-well plate (SPL, Republic of Korea) with grid lines until they had formed a confluent monolayer. Then, a scratch was made on the culture using a sterile plastic micropipette in order to simulate an in vivo wound. Ideally, the scratch had smooth edges, minimal cellular debris and a uniform width. After that, CTRL, 1.25% CB-PL and 0.3125% PB-PL were added, respectively. The snapshot method was performed, using Olympus Inverted Microscope Models IX71 to take sequential pictures of the simulated wound for 72 h with intervals of 24 h. The sequential pictures were taken from 3 spots based on the grid lines in each group (Figure 6a). After taking the cell microscopic image at 72 h/day 3, the cell culture media of each group were collected for PDGF-BB measurement, while the cells were treated with TRI reagent for qPCR. For data analysis, wound area was measured using Axiovision Version 18.4. Wound area was the cell-free area within the scratch. The microscopic image was calibrated with the 100× magnification and then used the outline feature in the Axiovision software was used to measure the surface area of wound (Figure 6b). The average of wound areas from 3 spots were calculated for each group.

The percentage of wound closure was calculated using the equation as follows:(2)$$\%\;\mathrm{Wound}\;\mathrm{Closure}=(\frac{A_{t=0h}-A_{t=∆h}}{A_{t=0h}})\times 100\%$$

*A_t=0_*_h_ = Area of wound measured immediately after scratching (*t* = 0 h);*A_t=_*_Δh_ = Area of wound measured at h hours after the scratch was performed.

The cell migration rate was calculated using the equation as follows:(3)$$\mathrm{Cell}\;\mathrm{Migration}\;\mathrm{Rate}=(\frac{A_{t=0h}-A_{t=∆h}}{\mathrm{day}})$$

*A_t=0_*_h_ = Area of wound measured immediately after scratching (*t* = 0 h);*A_t=_*_Δh_ = Area of wound measured at h hours after the scratch was performed.

### 4.4. Two-Step Quantitative Polymerase Chain Reaction (qPCR)

The total ribonucleic acid (RNA) of HOMFs were extracted using TRI reagent (Molecular Research Center, Cincinnati, USA), according to the manufacturer’s protocol. Briefly, the 1 mL cell lysate was separated into the aqueous and organic phases with the addition of 200 μL chloroform and centrifugation at 12,000 rpm for 15 min. RNA remained exclusively in the aqueous phase and precipitated through 500 μL of isopropanol and 5 μL of a polyacryl carrier (Molecular Research Center, Cincinnati, OH, USA). The precipitated RNA was washed with 1 mL 75% ethanol and solubilized with 10 μL RNAse and DNAse free distilled water (Invitrogen, Carlsbad, CA, USA). The quantity and purification of RNA was determined using a Nanodrop ND-100 spectrophotometer (Nanodrop Technologies, Wilmington, Delaware, DE, USA). A SuperScript™ IV First-Strand Synthesis SuperMix (Invitrogen, USA) was used to synthesize complementary deoxyribonucleic acids from 100 ng of total RNA, according to the manufacturer’s protocol. The reverse transcription was initiated with 10 min at 23 °C for primer annealing, 60 min at 50 °C for reverse transcription and 5 min at 85 °C for reaction termination. The primers (forward and reverse) were designed from NIH GenBank using Primer Output 3 software. GIyceraIdehyde-3-phosphate dehydrogenase (GAPDH) was used as a housekeeping gene for data normalization. There four target genes were Col. I, Col. III, elastin and fibronectin. The primer sequences of ECM markers and GAPDH were shown in Table 1 [32]. qPCR was carried out using Luna^®^ Universal qPCR master mix (NEB, Ipswich, MA, USA) in Bio-Rad iCycler (Bio-Rad, Hercules, CA, USA). The protocol condition was initiated with the activation of Taq DNA polymerase at 95 °C for 3 min, followed by 40 cycles of PCR amplification at 95 °C for 10 s and 61 °C for 30 s and then melt curve analysis.

### 4.5. Enzyme-Linked Immunosorbent Assay (ELISA) for PDGF-BB Measurement

ELISA kit (Biorbyt, Cambridge, UK) was used to quantify the protein level of PDGF-BB, according to the manufacturer’s protocol. The microplates are coated with the capture antibodies which were specific for human PDGF-BB protein. Standards and cell culture supernatant (day 3) of three groups (CTRL, CB-PL and PB-PL) with the volume of 100 µL were pipetted into the wells. During the first 90 min incubation with gently shaking at room temperature, the antigen bound to the capture antibody. After washing, the biotinylated anti-human PDGF-BB antibody was added to the wells and this antibody bound to the immobilized protein captured during the first incubation. After the removal of the unbound biotinylated antibody, a Horseradish Peroxidase-conjugated streptavidin was added to the wells. After that, a TMB substrate solution was added and colors developed in proportion to the amount of PDGF-BB bound. The stop solution changes the color from blue to yellow. The intensity of the colored product was read using ELISA microplate reader at 450 nm wavelength.

### 4.6. Statistical Analysis

The statistical analysis of data was performed using Statistical Package for the Social Sciences (SPSS) version 28.0 software (IBM Corp, Armonk, New York, NY, USA). Parametric tests involving one-way analysis of variance (ANOVA) were performed to assess the differences between group means and post hoc Tukey HSD tests were used to test differences among sample means for significance. The independent t-tests were used to compare the means of two groups. Non-parametric tests involving Kruskal–Wallis tests were used to determine the differences of more than two groups, where Mann–Whitney U tests were used to compare differences between two independent groups. The statistical differences were considered significant if *p* < 0.05.

## 5. Conclusions

In summary, we concluded that PB-PL is more effective than CB-PL in promoting cell proliferation and wound-healing as the optimal concentration of PB-PL (0.3125%) is lower than CB-PL (1.25%). This could be due to the higher amount of PDGF-BB in PB-PL compared to CB-PL. Nevertheless, PL from both sources can be a beneficial treatment for wound-healing. CB-PL can also enhance wound-healing, thus enabling it to be an alternative source of blood for wound treatment. However, PB-PL showed the most promising wound-healing properties in this study. The limitation of our study was the identification of limited biomarkers in wound-healing process was not performed. The detection of more biomarkers as well as their pathways are important with the purpose to determine factors that influence the difference in the wound-healing rate between PB-PL and CB-PL. A precise mechanism of high cell proliferation rate promoted by certain growth factors of PL has yet to be explored. Additionally, the effective use of CB-PL in a population of patients with frequent or recalcitrant oral ulcerations is next to be investigated.

## Figures and Tables

**Figure 1 ijms-24-05775-f001:**
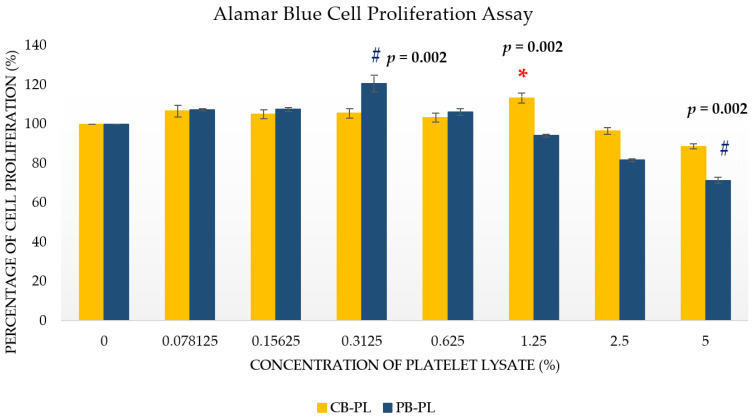
The percentage of cell proliferation for CB-PL and PB-PL were calculated vs. the serum-free medium/CTRL (normalized as 100%). All data were representative of six independent tests with *n* = 6 by groups and means ± SEM. * *p* < 0.05 denoted the statistical significance of CB-PL concentrations when CTRL (0% PL). ^#^
*p* < 0.05 denoted the statistical significance of PB-PL concentrations when compared to CTRL (0%).

**Figure 2 ijms-24-05775-f002:**
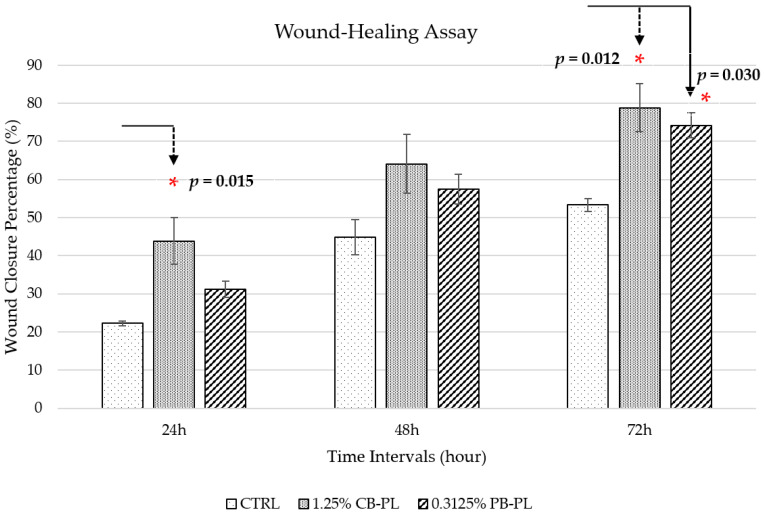
Wound-healing percentage of CTRL, 1.25% CB-PL and 0.3125% PB-PL towards the migration of HOMF. The wound closure was assessed after 24, 48 and 72 h of exposure to culture medium. All data were representative of three independent tests with *n* = 3 by groups and means ± SEM. * *p* < 0.05 denoted the statistical significance of 1.25% CB-PL and 0.3125% PB-PL when compared to CTRL.

**Figure 3 ijms-24-05775-f003:**
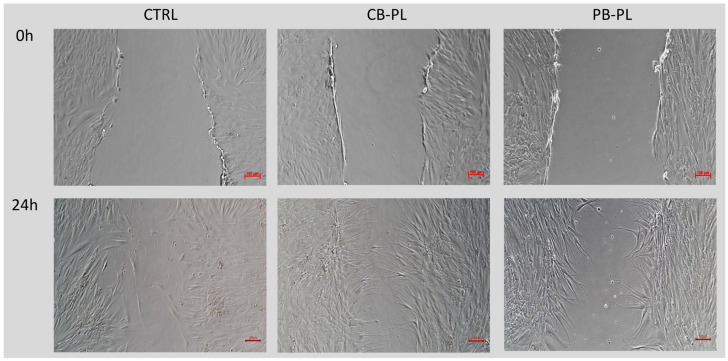
Phase contrast micrographs of scratch test showed the effect of CTRL, 1.25% CB-PL and 0.3125% PB-PL towards the migration of HOMF under 100× magnification. The wound closure was assessed after 24, 48 and 72 h of exposure to culture medium. The wounds were closed faster in 1.25% CB-PL and 0.3125% PB-PL compared to the CTRL.

**Figure 4 ijms-24-05775-f004:**
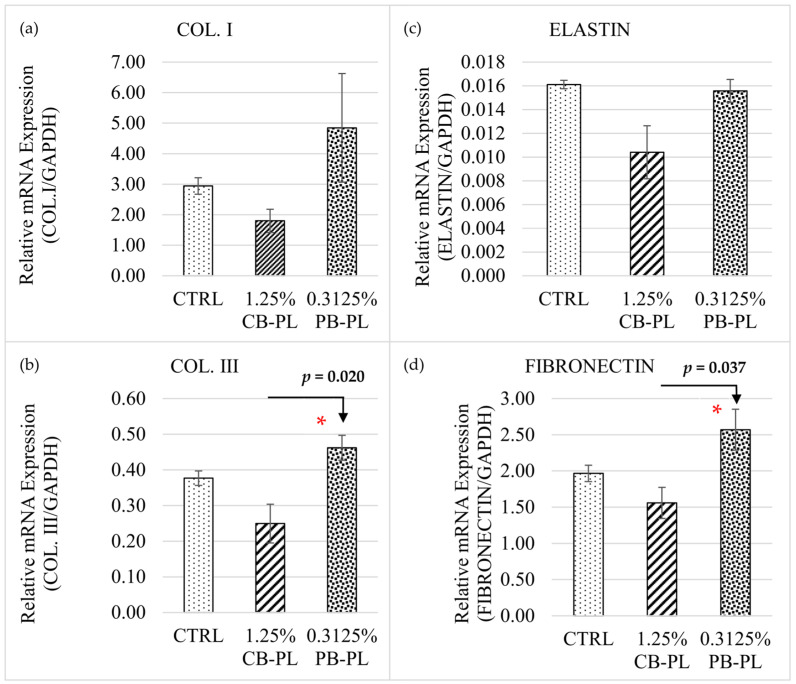
Relative mRNA expression of cell phenotypic markers (**a**) COL. I; (**b**) COL. III; (**c**) Elastin and (**d**) Fibronectin among CTRL, 1.25% CBPL and 0.3125% PB-PL groups on day 3. All data were representative of three independent tests with *n* = 3 by groups and means ± SEM. * *p* < 0.05 denoted the statistical significance of 0.3125% PB-PL when compared to 1.25% CB-PL.

**Figure 5 ijms-24-05775-f005:**
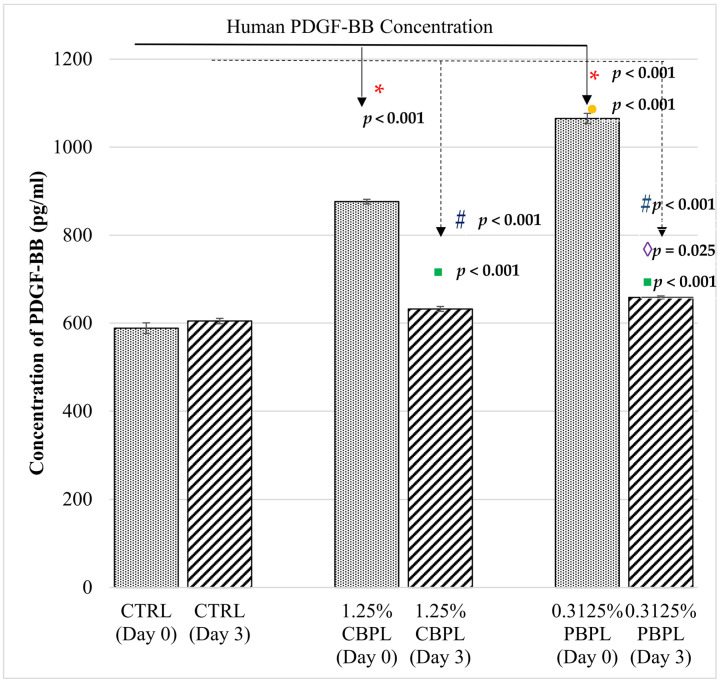
Human PDGF concentrations among CTRL, 1.25% CB-PL and 0.3125% PB-PL groups in a wound-healing assay on day 0 and 3. All data were representative of three independent tests with *n* = 3 by groups and means ± SEM. *^,#^
*p* < 0.05 denoted the statistical significance of 1.25% CB-PL and 0.3125% PB-PL when compared to control group on day 0 and day 3, respectively. ^●,◊^
*p* < 0.05 denoted the statistical significance of 0.3125% PB-PL when compared to 1.25% CB-PL on day 0 and day 3, respectively. ▪ *p* < 0.05 denoted the statistical significance of groups when compared day 3 to day 0.

**Figure 6 ijms-24-05775-f006:**
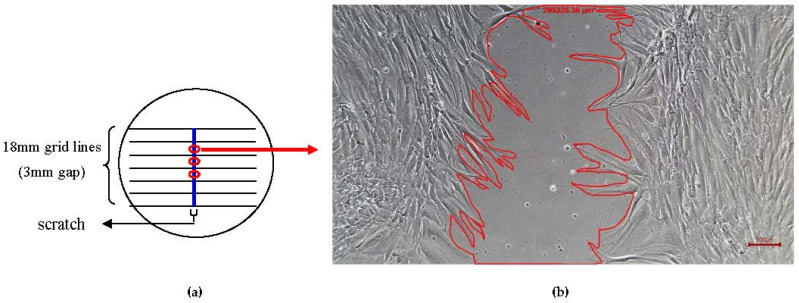
Scratch was made on the confluent monolayer cells to mimic a wound. (**a**) The microscopic images were taken from 3 spots sequentially for 72 h with intervals of 24 h in each group. (**b**) Wound area was measured using Axiovision Version 18.4. The wound area was represented by the cell-free area within the scratch and measured in unit of µm^2^. This microscopic image was taken after treatment with PB-PL at 24 h.

**Table 1 ijms-24-05775-t001:** Primer sequences of ECM markers and housekeeping gene used in this study.

Gene	GenBank Accession Number	Primer Sequence (5′ to 3′)
GAPDH	NM_002046.5	F: CAATGACCCCTTCATTGACC
		R: TTGATTTTGGAGGGATCTCG
Col. I	NM_000088.3	F: GTGCTAAAGGTGCCAATGGT
		R: ACCAGGTTCACCGCTGTTAC
Col. III	NM_000090.3	F: CCAGGAGCTAACGGTCTCAG
		R: CAGGGTTTCCATCTCTTCCA
Elastin	NM_000501.4	F: GGTGGCTTAGGAGTGTCTGC
		R: CCAGCAAAAGCTCCACCTAC
Fibronectin	NM_212482.2	F: AAAATGGCCAGATGATGAGC
		R: TGGCACCGAGATATTCCTTC

## Data Availability

Not applicable.

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
