# Peer review of "In Vitro Cell Proliferation and Migration Properties of Oral Mucosal Fibroblasts: A Comparative Study on the Effects of Cord Blood- and Peripheral Blood-Platelet Lysate"

_ijms, 2023, doi:10.3390/ijms24065775_

Round 1

Reviewer 1 Report

In this work, the Authors aimed to compare the effectiveness of Cord blood-platelet lysate (CB-PL) and peripheral blood-platelet lysate (PB-PL) in promoting oral wound closure in vitro. In particular, the objectives of this study are to determine the appropriate concentrations of CB-PL and PB-PL in wound healing, as well as to compare the healing rates and properties of CB-PL and PB-PL. The topic is relevant in the field of tissue regeneration and a consistent number of works about the use of cord blood and peripheral blood platelet lysate in wound healing are already present in literature. The work is well written and described but it lacks of important characterization to confirm that the use of PB-PL is better than the use of CB-PL. Overall, the study showed that PB-PL contained higher concentration of col I, coll III, fibronectin and elastin than CB-PL only from a molecular point of view. This is also be discussed by the Authors confirming that the limitation of the study was the identification of limited biomarkers in wound healing process.

In this context, I would like to ask to the Authors the characterization of the PL from CB and PB, for example using mass spectrometry. Please refers to https://doi.org/10.1177/2041731419845852 for PL characterization.

Also, the Authors didn’t show the total platelet concentration.

In addition, the Authors didn’t perform a cells phenotype characterization. Only a relative mRNA expression have been performed.

They show the percentage of proliferation but it is not clear the cells doubling time during wound healing.

Do the Authors think should be interested analyzing the differentiation capability of the cells after treatment with PL from different sources?

Figures 2, 4, 5: it is difficult to understand the statistical differences between the conditions tested.

Reviewer 2 Report

Dear Editor,

The manuscript “In Vitro Cell Proliferation and Migration Properties of Oral 2 Mucosal Fibroblasts: A Comparative Study on the Effects of 3 Cord Blood- and Peripheral Blood-Platelet Lysate ” describes a comparative study of the use of platelet lysates from the cord and peripheral blood for wound healing. The manuscript is written logically and in understandable English. However, the presented text in my opinion contains information that can be further explained, specified, and clarified. I have the following comments on the submitted manuscript:

INTRODUCTION

·        the authors state here that "Many researches had shown that PL can be used as an adjunct treatment for wound  healing. ", I recommend adding relevant specific references, at least of the most fundamental studies.

MATERIALS and METHODS

·        wound healing assay: the methodology for measuring/calculating the wound area is not described. How was the area calculated? From microscopic images? How?

RESULTS

·        In general: I lack a comparison with the complete media (containing FBS). It stands to reason that when using a serum-free control (minimal medium) it will always be worse compared to a PL-supplemented medium. However, have the PLs positive effect even when compared to the complete medium? If the experiment was not carried out, I recommend citing data from the literature and justifying the use of minimal medium as a control.

·        very nice images from the wound healing experiment. I recommend adding how the area after healing was determined/calculated, it is not clear from the methodology and in my opinion it is not a commonly known methodology.

·        PDGF has been measured in all samples CB and PB PL-supplemented and control (serum-free minimal medium) samples. Please explain the reason for the presence of analyzed PDGF in this control. What source does it come from?

DISCUSSION

·        I am missing a discussion regarding the response of the cells used to the tested concentrations of PL. Why, for example, is there no rising trend, but we always only see an increase in proliferation at a single concentration? On the contrary, with both (but more pronounced with PB-PL) we see inhibition of cell proliferation from a certain concentration. How do you explain that? It is possible to relate this to the concentration of proteins present and possibly compare it to the proteins in the complete medium (10% FBS is commonly used here).

·        how do authors explain that the use of CB_PL is optimal at a higher concentration, while PB_PL is at a lower .... you also mention that the literature indicates that CB contains a higher amount of GFs than PB - so what is the reason?

·        authors write "Both PB-PL and CB-PL showed a higher rate of wound healing compared  to the control", but what would the situation look like if complete media were used? I recommend adding data from the literature here.

CONCLUSIONS

This part should be improved on the basis of improved results and discussion. E.g. the authors claim "we found that CB-PL can be as effective as PB-PL in 519 wound healing", however, I consider this misleading, as PB-PL was used in a different (lower) concentration than CB-PL.

For the above reasons, I propose the revision of the article. 

Reviewer 3 Report

1- Statistics on all graphs should be demonstrated illustrating p-value

2- Resolution of wound scratch figure needs improvement. The scale bar is very hazy and of low resolution.

3- Coloured ligands should be shown on all figures instead of letters a and b.

4- Conclusion is very superficial. It needs improvement and more details with numerical results should be displayed.

5-All graphs should be constructed using the same software. Please change proliferation graph to be similar to wound healing assay.

Round 2

Reviewer 1 Report

The manuscript can be accepted in present form.

Reviewer 2 Report

Dear editor,

All my comments have been taken into account in the revised version of the manuscript “In Vitro Cell Proliferation and Migration Properties of Oral 2 Mucosal Fibroblasts: A Comparative Study on the Effects of 3 Cord Blood- and Peripheral Blood-Platelet Lysate ” and I have no further comments.